# Remote Heart Rate Prediction in Virtual Reality Head-Mounted Displays Using Machine Learning Techniques

**DOI:** 10.3390/s22239486

**Published:** 2022-12-05

**Authors:** Tiago Palma Pagano, Lucas Lisboa dos Santos, Victor Rocha Santos, Paulo H. Miranda Sá, Yasmin da Silva Bonfim, José Vinicius Dantas Paranhos, Lucas Lemos Ortega, Lian F. Santana Nascimento, Alexandre Santos, Maikel Maciel Rönnau, Ingrid Winkler, Erick G. Sperandio Nascimento

**Affiliations:** 1Computational Modeling Department, SENAI CIMATEC University Center, Salvador 41650-010, Bahia, Brazil; 2HP Inc. Brazil R&D, Porto Alegre 90619-900, Rio Grande do Sul, Brazil; 3Department of Management and Industrial Technology, SENAI CIMATEC University Center, Salvador 41650-010, Bahia, Brazil; 4Faculty of Engineering and Physical Sciences, School of Computer Science and Electronic Engineering, Surrey Institute for People-Centred AI, University of Surrey, Guildford GU2 7XH, UK

**Keywords:** artificial intelligence, head-mounted displays, heart rate, neural network, regions of interest, machine learning, virtual reality

## Abstract

Head-mounted displays are virtual reality devices that may be equipped with sensors and cameras to measure a patient’s heart rate through facial regions. Heart rate is an essential body signal that can be used to remotely monitor users in a variety of situations. There is currently no study that predicts heart rate using only highlighted facial regions; thus, an adaptation is required for beats per minute predictions. Likewise, there are no datasets containing only the eye and lower face regions, necessitating the development of a simulation mechanism. This work aims to remotely estimate heart rate from facial regions that can be captured by the cameras of a head-mounted display using state-of-the-art EVM-CNN and Meta-rPPG techniques. We developed a region of interest extractor to simulate a dataset from a head-mounted display device using stabilizer and video magnification techniques. Then, we combined support vector machine and FaceMash to determine the regions of interest and adapted photoplethysmography and beats per minute signal predictions to work with the other techniques. We observed an improvement of 188.88% for the EVM and 55.93% for the Meta-rPPG. In addition, both models were able to predict heart rate using only facial regions as input. Moreover, the adapted technique Meta-rPPG outperformed the original work, whereas the EVM adaptation produced comparable results for the photoplethysmography signal.

## 1. Introduction

The estimation of heart rate is essential for monitoring humans in a variety of situations, such as driving a vehicle [1], practicing physical activities [2], working under hazardous conditions [3] or during an investigative police interrogation [4]. The variability of heart rate can be used to map and identify stress [1], fatigue [5], emotions [6], health [7] and social behavior [8] indicators.

Heart rate is a vital body signal that enables the monitoring of a person’s health. In addition to conventional techniques of measurement, visual estimation typically employs a camera or a low-cost, minimally invasive device [9]. In recent years, research on heart rate estimation by video has increased [10,11], while PhotoPlethysmoGraphy (PPG), a technique for measuring variations in blood volume on a variety of devices to estimate heart rate [12] has gained popularity.

Heart rate is the number of times the heart makes blood pulse in one minute [13]. In general, for an adult during rest, heart rate commonly ranges between 60 and 100 Beat Per Minute (BPM) [13,14,15]. The electrocardiogram is a test that measures the resting heartbeat rhythm [16] and can be used to diagnose a patient’s heart health. PPG, on the other hand, is commonly obtained by wrist or finger oximeter.

The estimation of heart rate from facial videos is being refined to enable non-invasive monitoring of cardiac information [9] and to simplify capture devices by eliminating the need for heart rate sensors, requiring only cameras that capture the user’s face [9]. Numerous devices have cameras to capture the user’s face, including virtual and augmented reality devices, which use facial features and expressions to enhance the user’s immersion in the virtual environment. Ergo, these cameras may be employed to estimate the user’s heart rate and monitor their condition in a particular context.

It is possible to determine the heart rate from facial videos using Machine Learning (ML) [17,18,19]. ML enables the recognition of patterns in data containing multiple variables with discrete variations that have a significant impact on the outcomes [20]. Particularly, deep learning techniques consist of neural networks with multiple layers and high levels of flexibility, allowing for the classification of various cases in an efficient and effective manner. Consequently, the identification of heart rate from video can be generalized by adjusting the internal parameters to represent the desired dataset structure and by identifying the optimal combinations of complex feature data.

This functionality enables the creation of autonomous systems capable of human-like decision making. Even nowadays, acquiring suitable data to train a neural network is a challenge for many problems in the fields of object detection and recognition, computer vision, speech recognition, natural language processing, social computing processing and sentiment analysis, as stated in [21]. ML is widely used in healthcare, as in [22], where the model is used to predict the outcomes of interactions between osteoporosis drugs and Paget treatment by extracting chemical features that interact in previously defined drug pairs. In the work of [23], a prediction model using ML is proposed, incorporating a genetic algorithm to predict low-grade glioma molecular subtypes using magnetic resonance imaging. There are also works that support COVID-19 identification from chest computed tomography scans, such as [24], which proposes a deep learning system inspired by the Inception V3 architecture and [25], which presents a light 3D Convolutional Neural Network (CNN) that supports COVID-19 detection from chest X-ray scans. Both of these support the diagnosis of COVID-19. All of these studies demonstrate that ML models can accurately predict elements of human health and physiology.

Head Mounted Display (HMD) are virtual reality devices that occasionally feature cameras in the eye region and the underside of the face. The absence of a dataset containing only the eye and lower face regions necessitates the development of a method for simulating videos produced by an HMD by extracting the regions from the existing dataset. Similarly, there is no related work for BPM prediction that utilizes only highlighted facial regions; therefore, an adaptation is required for BPM predictions.

In this context, the objective of this work is to remotely estimate heart rate from facial regions that can be captured by the cameras of a head-mounted display using state-of-the-art EVM-CNN [26] and Meta-rPPG [27] techniques. The source code used for this work is available at Github [28].

The EVM-CNN [26] technique estimates heart rate remotely under realistic conditions, excluding the eyes and mouth to identify a more stable region and defining the central portion of the face as the primary area. The Meta-rPPG [27] technique estimates an individual’s heart rate remotely from a video camera using facial landmarks.

This paper is organized as follows: Section 2 defines research-relevant concepts, Section 3 describes the experiment’s methods, Section 4 presents the findings and discusses the highlights and Section 5 offers the conclusions and suggestions for future research.

## 2. Background

The research-relevant concepts are presented in the following sections.

### 2.1. Datasets

The following are summaries of the datasets that best meet the criteria for measuring heart rate.

The public dataset MR-Nirp [29] is divided into two contexts, inside a car and inside a room. Both datasets have video recorded in RGB and near-infrared. The MR-Nirp Car contains 180 face videos of 18 participants in a car, i.e., 10 experiments per participant. The MR-NIRP-INDOOR is recorded in a room, the face of each of the eight participants is filmed either still or moving. The contexts have files with data from a pulse oximeter attached to the participants’ finger. The age of the participants is between 20 and 60 years old, containing two women and 16 men, in which four of them have a beard. The instructions for use of the dataset put that skin tones vary between Indian, Caucasian and Asian. The dataset does not require a request nor consent for academic use.

The public dataset UBFC-rPPG [30] contains 42 videos of the participants’ faces, and the database has PPGs and BPMs produced with a pulse oximeter and with frames of 640 × 480 pixels. The dataset is focused for Remote PhotoPlethysmoGraphy (rPPG) analysis, and no subscription term is required to obtain it.

### 2.2. EVM-CNN Technique

The indication of the region of interest selected for the application of the EVM-CNN [26]. Once this region is identified, the image is spatially decomposed into multiple bands using the Gaussian pyramid, the band that has the lowest frequency is remodeled from 2D to 1D. After this process, the reshaped images are concatenated to obtain a single image. Each column of the result or feature image is composed of the reshaped images of the region of interest, and each row corresponds to the color variations of one pixel at a fixed position in the region of interest. The channels of the new image are changed to a new frequency domain (0.75 to 4 Hz) by applying the Fast Fourier Transform (FFT). These channels are also multiplied with a mask to zero out components that are outside the frequency range specified in the FFT. Finally, the channels are transferred to the frequency domain prior to the application of the FFT and connected again to obtain the feature image. The resulting image serves as the input of the regression-oriented CNN inspired by the MobileNet network.

The architecture of the Eulerian Video Magnification (EVM) [26] consists of using a regression CNN. It uses the generated feature image as input to the model, the expected output consists of the heart rate value. Depth-separated convolutions were used as the main structure of the network, which can be seen in Table 1, inspired by the architecture of the MobileNet network. This form of convolution was employed in order to factor and transform into deep, as well as to aid in reducing the computational load and model size.

The first layer of the model is composed of a full convolution, however, the second layer of the network has a deep convolution to filter the input channels. In addition, a point layer is used to gather the outputs coming from the previous layer. These calls are named “Dw” and “Pw”. The applied kernel has the size 5 × 5 × 3 × 96, where 5 × 5 is the size of the filter height and width, 3 is the value for the input channel RGB and 96 is the number of mapped output features. It also has hidden parameters, which are always equal to the value one, due to the convolution property in depth. After the convolution layer of the model was employed, the batch normalization technique and the ReLU activation function were also used. The averaging pooling layers were used to reduce the map resolution to value one and calculate the averaged values for each channel.

In order to improve computational performance and reduce the occurrence of overfitting in the model training, input data resolution reductions were performed throughout the network. Thus, a dropout layer was used at the end of the net to reduce overfitting and improve generalization ability. The output consists of only one neuron, since heart rate is only one label, since the model’s input feature image contains one second of information. Furthermore, all labels were normalized in the range of zero and one before the training step was started. The smallest of the beat range, 45 BPM, goes to the value zero, and the largest of the range, 240 BPM, goes to the value one. For the loss function, the Euclidean distance calculation was applied to calculate the difference between the values predicted by the model and the true absolute values.

### 2.3. Meta-rPPG Technique

In the Meta-rPPG [27] technique, after detecting the facial landmarks, the values of pixels that are in the identified regions are kept, but pixels that are outside this area have their values reduced to zero. After this, the resulting region of interest is cropped and reshaped to a new size T × T. Then, the facial images are coupled to the PPG signal and extracted using a PPG sensor located on the participant’s finger. To perform this estimation, the authors presented the application of learning by meta-learning, which promotes a faster adaptation of the algorithm to small changes.

This learning was employed in a deep neural network divided into two parts, a CNN for feature extraction and an Long Short-Term Memory (LSTM) network for rPPG estimation. The architecture of the convolutional encoder was based on the ResNet network, and from the temporal information, it models the visual information of the input images. In addition, the modeling of the PPG signal is performed by applying the LSTM rPPG estimator, which separates the rPPG signal from the visual modeling. The name “Meta-rPPG” [27] comes from the inference of rPPG using a meta-learner, which is therefore the use of a model that adapts quickly from a few examples of the new context, using a technique called few-shot. The few-shot feature of Meta-rPPG defines the regression problem using part of the input sequences of the network as support, as pre-training for adaptation, partly as query and as performance evaluation of adaptive learning. In addition, a synthetic gradient generator modeled by the Hourglass network was used to assist in transduction inference, it performs faster adaptation with unlabeled data minimizing the distance between out-of-distribution samples and the samples used to train the network.

The network structure is composed of Conv2DBlocks, with the Conv2D layer, Batchnorm, Average Pooling and ReLU activation function as the main components. In addition, shortcut connections are employed between Conv2DBlocks. The 1D convolution blocks consist of the Conv1D layer, Batchnorm and the ReLU function. The ✓ symbol in the Table 2 indicates which category of information the layer is acting on.

## 3. Materials and Methods

The methodology for this work involves achieving two goals: extracting the Region of Interest (ROI) and adapting related work for BPM prediction based on ROIs alone, with this, the heart rate prediction is relayed according to the diagram in Figure 1.

### 3.1. Region of Interest Extractor

The construction of the algorithm aimed at clipping the regions of interest from the videos so that they would resemble the data from the cameras of an HMD. In addition, the technique of threshold was used as a stabilizer for the videos processed by the algorithm. The use of threshold also enabled less occurrence of unwanted motion, providing more stable images to be processed, without adding other problems in the cropped area.

Upon starting the algorithm, a stream will be run through each frame of the original video, with the previous frame removed from memory to avoid an overload of the system. After the frame is loaded, the facial landmarks are detected with either FaceMesh or the Support Vector Machine (SVM), shown in Figure 2.

The ROIs of the face of the individual in the video are represented in the Figure 2, the blue square refers to the ROI of the right eye, the red to the left eye and the green to the mouth region. The yellow rectangle selects the region in the middle of the face that starts from the bottom of the eyes to the top of the mouth; this region was not used in the experiments of the present work and only serves to represent the ROI that the EVM authors use in [26]. During the detection phase of the ROIs, in case of error, the algorithm keeps the last detected coordinates, following the normal flow of execution, except when the error occurs consecutively, causing the interruption of the video processing.

The detector receives the frame and predicts the facial points, where each point is composed of two values indicating its coordinate in the image. The center point of each ROI is calculated from the average of the coordinates of the points in the region. With *C* being the list of points in an ROI, $x_{i}$ and $y_{i}$ being its coordinates and *N* being its total size, we can represent the center point $\bar{C}$ of an ROI as in Equation (Equation 1).
(1)$$\bar{C}=\left[\frac{\sum_{i=1}^{N}x_{i}}{N},\frac{\sum_{i=1}^{N}y_{i}}{N}\right]$$

The Euclidean distance calculation is performed with ${\bar{C}}_{previous}$ in Equation (Equation 2) and ${\bar{C}}_{current}$ in Equation (Equation 3), corresponding to the center points of the same ROI.
(2)$${\bar{C}}_{previous}=\left[x_{previous},y_{previous}\right]$$
(3)$${\bar{C}}_{current}=\left[x_{current},y_{current}\right]$$

The greater the calculated distance, the greater the evidence of face motion, which should cause the ROI to move in the same direction. The smaller the calculated distance, the greater the evidence of stability in the movement of the face. However, the distance is not zeroed and does not remain constant, due to natural movement of the face, camera shake, or variation in the prediction of facial landmarks. Therefore, even if the face remains stationary, the cropped ROIs will suffer shaking if stabilization is not applied.

After determining the center point $\bar{C}$ of the ROI, the algorithm uses half the length of the ROI to create a square of center, and $\bar{C}$ is used to crop the frame in Figure 3. An approximation distortion in the mouth region is applied after its cropping, considering a radius between the mouth and the nose, obtaining a center, so that the result looks like the face is closer to the camera. Figure 3 shows the ROI of the mouth without the distortion and the same region with the distortion. When cropped, the ROIs are resized to the final size of the eyes and mouth, to 100 × 100 and 400 × 400, respectively.

The position that the frame occupies in the video is registered in the name of the saved file of type Portable Gray Image when the image is captured with an infrared sensor, and as Portable Network Graphics when the image is captured with a visible light sensor.

To define when the coordinates of the ROI should be updated, the calculated Euclidean distance *d* can be submitted to a threshold to determine the behavior of the algorithm when the distance exceeds the threshold or not. If the distance is less than the threshold, the last points of the ROI are kept, ignoring the new detection that would cause a jitter. If the distance is greater than the threshold, the new detected points will be used to move the ROI according to the movement of the face. The behavior of the algorithm should also consider the distance from the camera, since, for example, the movement of a face five meters away results in a smaller Euclidean distance than the movement of the face of someone who is half a meter away from the camera. So, a fixed threshold would not have the same quality and, to consider the distance of the subject from the camera, the threshold is calculated at 5% of the ROI length.

There are two actions to take when changing face landmarks when the threshold is exceeded. The first uses an average between the old and the new detection points, resulting in a smooth transition. The second action replaces the new detection with the previous one without the average, resulting in a less smooth transition.

The video magnification [31] was applied in the extractor of the ROI, an optional parameter, in order to highlight subtle movements in the videos that will be used after extraction. Amplification of periodic color variations was not used in the videos of our experiments. The motion amplification used is based on the present frames, without having the influence of future frames. This technique follows the Eulerian approach, which consists of three states: the decomposition of frames into an alternative representation, the manipulation of it and the manipulated reconstruction for the magnified frames. The decomposition of the frames results in edge-free videos with better amplification characteristics.

### 3.2. Adaptation of Techniques from Related Work to Work with the Regions of Interest

The implementation of the adaptation of EVM-CNN was divided into two steps: extraction of the features and processing through the adapted MobileNet network. The technique of feature images proposed in [26] was reproduced based on the pseudocode and descriptions provided.

The adaptation of the MobileNet network was implemented based on the summary available in the EVM article [26]. In the absence of information on optimizer choice, the Adam algorithm was chosen. The only divergence from the proposed technique was the threshold technique over the Scalable Kernel Correlation Filter Tracking technique [32].

After implementing the EVM model, the first improvement was the change in the original prediction target, previously performed directly with BPM prediction to PPG prediction; this change was accomplished by the possibility of its conversion to BPM using HeartPy library, which handles heart rate analysis tasks that have noisy data. In addition, the videos used are rescaled to 30 FPS. Other than that, the adaptations made to the EVM enabled the prediction of the remote heart rate by clipping new ROIs, being: the right eye, the left eye and the lower part of the face.

The Meta-rPPG has a source code available on the GitHub platform. However, despite being promising, its training and testing were associated with a single dataset, making it exhaustive to make any changes. To make it work in different datasets, some modifications were made, the first in the data input and the second in the code scalability and separation of pre-training and training data.

The input is linked to the output of the regions of interest extractor, which provides three frames. One of them being from the lower face region and two from the eye regions. With this, upon receiving these frames, the code concatenates the frames of the eyes horizontally and resizes the frames of the lower face region with the frames of the eyes vertically, so that they are positioned according to the human face. Once they are all in one frame, a final resize is performed to the size of 64 × 64 pixels.

Another point of adaptation of the Meta-rPPG was the adjustment of the input values of the dataset data, previously static for a 60 s duration, then passed to dynamic, since the static values of the input imply data loss in training, since videos shorter than 60 s were discarded and videos longer than 60 s were reduced. In addition, the videos used are rescaled to 30 FPS.

For the training and testing steps, the dataset was split into 88% and 12%, respectively. The data separation is performed for the pre-training, where 22% of the training dataset is forwarded to the pre-training to be divided into 55% for query and 45% for support, because in the pre-training stage, the few-shot methodology is used for initial learning.

To demonstrate this diversity, tests were performed on two datasets, with the facial landmark detection technique FaceMesh. Three additional versions of the dataset UBFC-rPPG were also generated in order to obtain better results by applying video magnification, using the SVM facial landmark detection technique and changing the RGB channels to grayscale.

All assessment metrics are based on the error (E), measuring the difference between the predicted (P) and ground truth (GT) values, respectively: Mean Error (ME), Standard Deviation (SD), Root Mean Square Error (RMSE), Mean Absolute Percentage Error (MAPE) and Pearson’s correlation (ρ).

Error is measured as the difference between predicted values and ground truth values, as expressed in Equation (Equation 4).
(4)$$E_{i}=P_{i}-GT_{i}$$

The mean error is the average of all errors in a set. Therefore, a better answer for this metric is if the result is closer to zero, the equation can be seen in Equation (Equation 5).
(5)$$ME=\left(\frac{1}{n}\right)\sum_{i=1}^{n}E_{i}$$

The standard deviation is the average of how far each value is from the mean, that is, the average amount of variability in the dataset used. Therefore, an SD close to zero indicates that any given data are closer to the mean. So, the equation is expressed in Equation (Equation 6).
(6)$$SD=\sqrt{\left(\frac{1}{n}\right)\sum_{i=1}^{n}{\left(E_{i}-ME\right)}^{2}}$$

RMSE can be considered a standard deviation of predictions errors. So this metric is the average distance between the model’s predicted values and the actual values in the dataset, so if it is closer to zero, it is a better result. Thus, the equation is expressed in Equation (Equation 7).
(7)$$RMSE=\sqrt{\left(\frac{1}{n}\right)\sum_{i=1}^{n}{\left(P_{i}-GT_{i}\right)}^{2}}$$

MAPE measures the average of the predicted quantities and the actual quantities of each entry in a dataset. Thus, the closer to zero, the better the results, and the equation can be seen in Equation (Equation 8).
(8)$$MAPE=\left(\frac{1}{n}\right)\sum_{i=1}^{n}\frac{|E_{i}|}{GT_{i}}$$

Pearson’s correlation ranges from −1 to 1, so a higher value indicates a relationship between the variables. The equation can be seen below, in Equation (Equation 9).
(9)$$\rho =\frac{\sum_{i=1}^{n}\left(GT_{i}-\bar{GT}\right)\left(P_{i}-\bar{P}\right)}{\sqrt{\sum_{i=1}^{n}{\left(GT_{i}-\bar{GT}\right)}^{2}}\sqrt{\sum_{i=1}^{n}{\left(P_{i}-\bar{P}\right)}^{2}}}$$

## 4. Results and Discussions

The results are obtained with EVM and Meta-rPPG from the ROIs extractor and video magnification. The results obtained are shown below in the Table 3 and Table 4.

### 4.1. Region of Interest Extractor

During the tests with the extractor, the detection of facial landmarks with FaceMesh obtained better results, as the clipping and detection showed higher returns than SVM. The better results of FaceMesh over SVM can be seen in Table 4. The technique performed well, recognizing the ROIs and extracting them, resulting in the frames used in the models; an example can be seen in Figure 4.

After adding the video magnification [31] in the region of the interest extractor, some motions that were imperceptible were amplified, but this did not mean an aid to the models. This behavior can be seen in Table 4 when we compare the metrics obtained by the scalable Meta-rPPG technique on the dataset UBFC-VM, which uses a video magnification technique and the same technique on the original dataset UBFC. Some of these movements may be associated with blood flow, however, muscle movements not related to heartbeat may bring noise that impairs network learning and inference. The application of video magnification on a UBFC video can be seen in Figure 5.

### 4.2. Adaptation of State-of-the-Art Techniques

The resulting images were resized to a ribbon and can be seen in Figure 6. Furthermore, other modifications made to the network allowed for improved results, among them, the addition of EVM scalability, allowing the model to be trained with a larger amount of data, increasing its performance. These modifications provided an improvement in the network’s prediction responses. Furthermore, after adapting the target for PPG, the network improved significantly, resulting in a higher ρ, as presented in Table 3 were the bold numbers are the best results overall.

In addition, we used ME, SD, RMSE, MAPE and ρ to verify the performance of the model, where values in bold indicate the best values achieved for each metric. The results of EVM with UBFC, using PPG, presented an ρ superior to the other tests performed, especially when compared using BPM with an increment of 188.88% in ρ, as can be seen in the Table 3. Moreover, EVM with PPG using UBFC presented a lower value for the MAPE error, showing that the model has a good prediction system of greater accuracy and correlation. Furthermore, this model does not show great results for RMSE, SD or ME. However, EVM with MR-NIRP-INDOOR showed high results for RMSE and SD, but not so for ρ, with the second best result for this metric. The EVM with UBFC and without PPG achieves good results only for ME but performs poorly for ρ. It is also noticeable that the EVM technique with PPG has a good performance in predicting the BPM values, which are very close to the ground truth values, when observed in the Figure 7.

The adaptations and improvements made to Meta-rPPG allowed satisfactory results, as well as the possibility of using a more diverse quantity of datasets.

Six tests were performed using Meta-rPPG and the FaceMesh region extractor, which can be seen in Table 4. The first two used the original Meta-rPPG technique for the datasets MR-NIRP-INDOOR and UBFC-rPPG, in which it can be seen that the non-scalable Meta-rPPG worked better with the UBFC-rPPG dataset than with the MR-NIRP-INDOOR, where its ME was superior to 100%, which implies that, for this dataset, it has close to zero prediction accuracy.

The next tests were improvements that made the technique scalable. In the third test, with the dataset UBFC-rPPG, there is noticeable improvement in the results, especially in the ρ, with a 55.93% increase. In the fourth test, video magnification was applied to UBFC-rPPG, and in the fifth, replacing the FaceMesh region extractor with SVM, it is possible to observe that there was a reduction in the ρ by 39% when compared with the non-scalable Meta-rPPG. As for the sixth, the conversion of the RGB channels to grayscale was performed to achieve a 25.42% increase when compared with the non-scalable Meta-rPPG, which is still smaller than compared with the unmodified dataset.

The graph of the BPM and PPG prediction of one of the individuals present in the dataset UBFC-rPPG of the third test, which obtained the best results among those tested through the scalable Meta-rPPG technique, can be seen in Figure 8 and Figure 9.

The results obtained by the Meta-rPPG model are better than the EVM results, where ρ is 76.92% higher than the EVM, as can be seen in Table 3 and Table 4. In addition, it is noticeable that the Meta-rPPG model works better with RGB images than grayscale ones and outperformed the original technique in some metrics, such as SD, RMSE and ρ. The results are presented in Table 4 and Table 5.

## 5. Conclusions

Since there are no publicly available datasets of videos produced by an HMD device, it was necessary to develop a region of interest extractor to simulate these videos. Clipping facial regions from the videos successfully extracted the desired points; additionally, the threshold enabled less interference from unwanted movements in the videos, resulting in more stable images of the model and mitigating issues in the clipped areas. So, the developed extractor was effective at obtaining regions of interest from public datasets containing facial videos and vital signs of individuals.

EVM and Meta-rPPG techniques were modified to receive the regions of interest clipped by the extractor as input and output. The results of the Meta-rPPG model are superior to those of the EVM, where ρ is 76.92% higher than the EVM. In addition, the Meta-rPPG model performed better with RGB images than grayscale ones and outperformed the previously proposed technique in terms of certain metrics, including SD, RMSE and ρ.

As a suggestion for future research, we propose using other existing sensors in an HMD to predict BPM, as well as developing a model to predict an individual’s emotions based on their vital signs and using neural networks to assist in signal conversion, which can significantly improve the results.

## Figures and Tables

**Figure 1 sensors-22-09486-f001:**
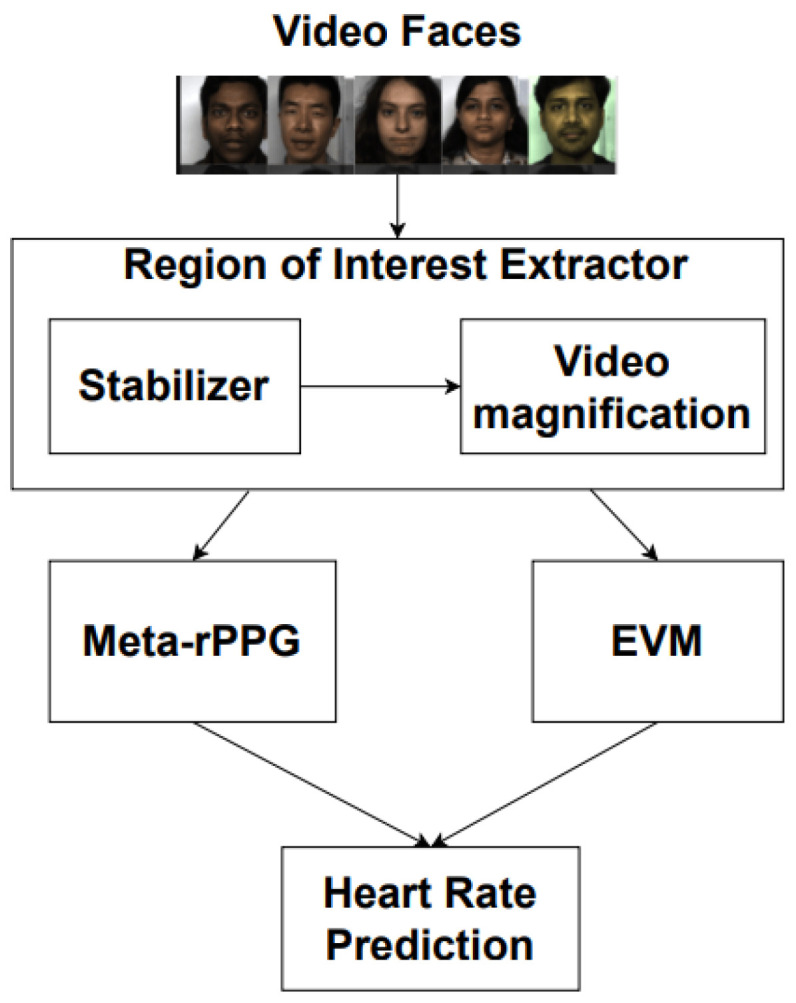
Methodology used.

**Figure 2 sensors-22-09486-f002:**
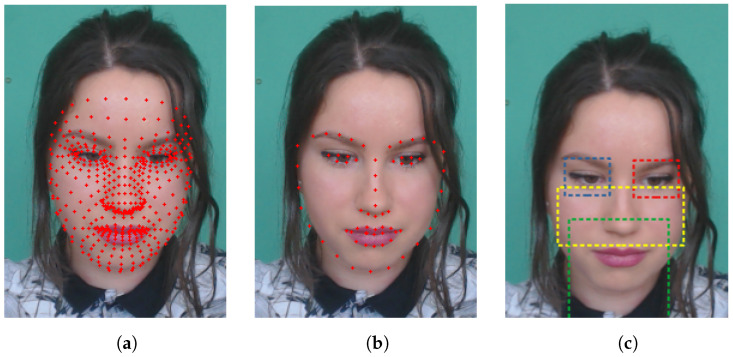
Images from the first subject of the dataset UBFC. (**a**) Facial features detected by FaceMesh. (**b**) Facial features detected by SVM. (**c**) Regions of interest that were selected.

**Figure 3 sensors-22-09486-f003:**
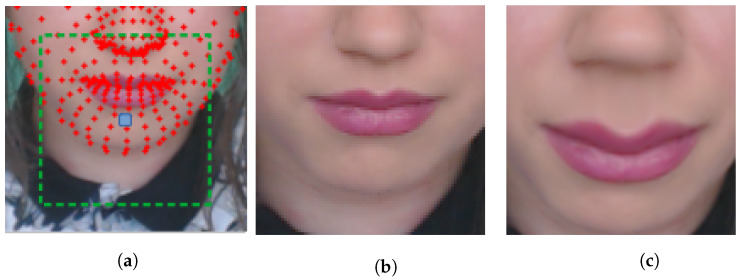
Exemplification of the construction of ROIs. (**a**) Example of the center point $\bar{C}$ of an ROI. (**b**) Example of the ROI cutout of the mouth without distortion. (**c**) Example of an ROI cutout of a mouth with distortion.

**Figure 4 sensors-22-09486-f004:**
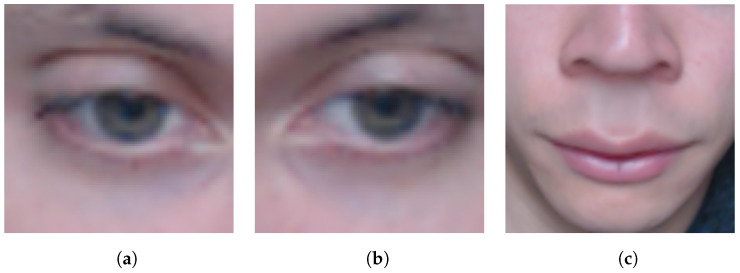
Exemplification of the frames of the ROIs. (**a**) ROI image of the right eye. (**b**) ROI image of the left eye. (**c**) Image of the lower face ROI.

**Figure 5 sensors-22-09486-f005:**
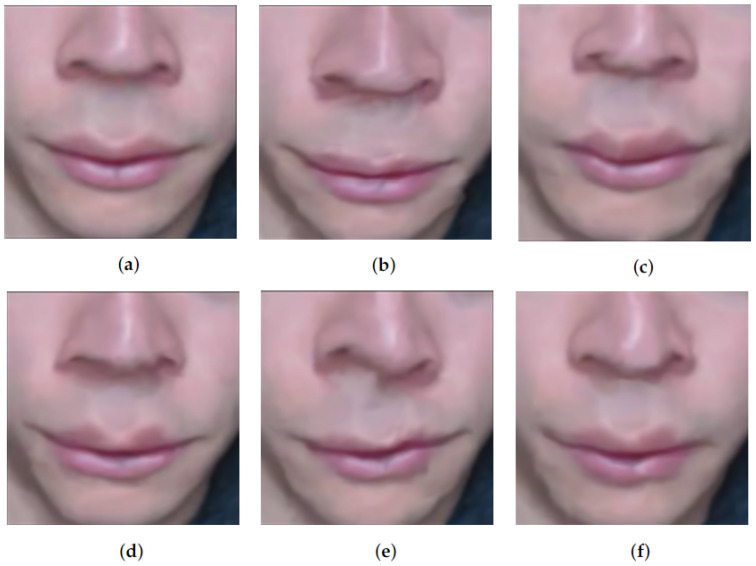
Bottom face frames after applying the video magnification technique. (**a**–**f**) sequence of frames with video magnification technique applied.

**Figure 6 sensors-22-09486-f006:**
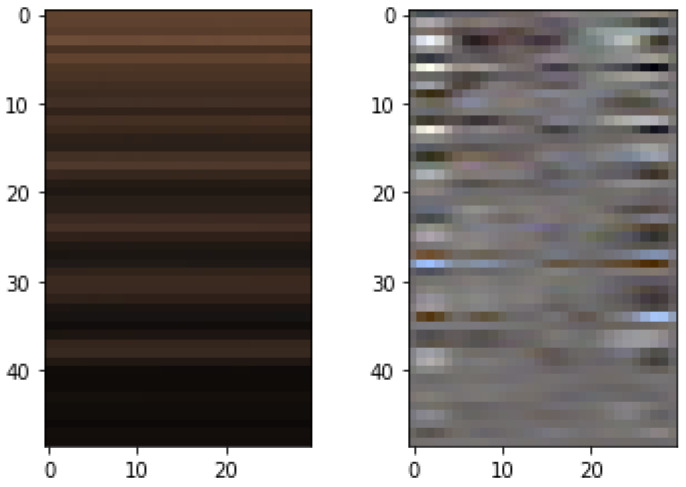
Feature image generated from the reproduction of the article [26].

**Figure 7 sensors-22-09486-f007:**
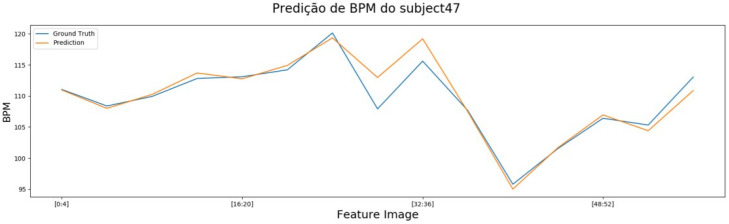
BPM chart with EVM results on subject 47 of the UBFC.

**Figure 8 sensors-22-09486-f008:**
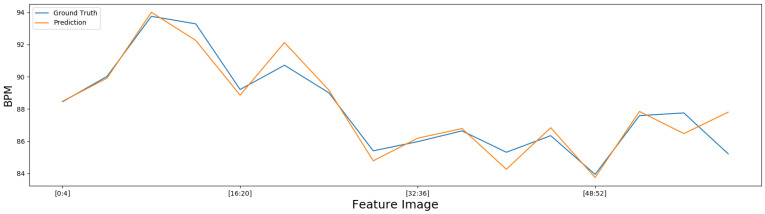
BPM prediction of subject 49 from dataset UBFC.

**Figure 9 sensors-22-09486-f009:**
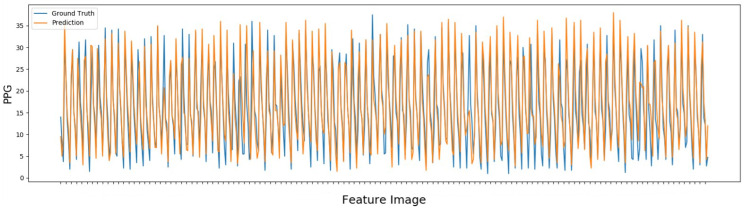
Prediction PPG of subject 49 of dataset UBFC.

**Table 1 sensors-22-09486-t001:** EVM-CNN Network Architecture.

Input Size	Type/Stride	Filter Shape
25 × 25 × 3	Conv/s1	5 × 5 × 3 × 96
23 × 23 × 96	DwConv/s1	3 × 3 × 96 dw
21 × 21 × 96	PwConv/s1	1 × 1 × 96 × 96
21 × 21 × 96	DwConv/s2	3 × 3 × 96 dw
11 × 11 × 96	PwConv/s1	1 × 1 × 96 × 96
11 × 11 × 96	DwConv/s2	3 × 3 × 96 dw
6 × 6 × 96	PwConv/s1	1 × 1 × 96 × 128
6 × 6 × 128	DwConv/s2	3 × 3 × 128 dw
3 × 3 × 128	PwConv/s1	1 × 1 × 128 × 128
3 × 3 × 128	DwConv/s2	3 × 3 × 128 dw
2 × 2 × 128	PwConv/s1	1 × 1 × 128 × 128
2 × 2 × 128	AvePool/s1	Pool 2 × 2
1 × 1 × 192	FC/s1	128 × 192
1 × 1 × 192	Dropout/s1	ratio 0.6
1 × 1 × 192	FC/s1	192 × 1
1 × 1 × 1	Eu/s1	Regression

Source: [26].

**Table 2 sensors-22-09486-t002:** Meta-rPPG network architecture.

Module	Layer	Output Size	Kernel Size	Spatial	Temporal
Convolutional Encoder	Conv2DBlock	60 × 32 × 32 × 32	3 × 3	✓	
	Conv2DBlock	60 × 48 × 16 × 16	3 × 3	✓	
	Conv2DBlock	60 × 64 × 8 × 8	3 × 3	✓	
	Conv2DBlock	60 × 80 × 4 × 4	3 × 3	✓	
	Conv2DBlock	60 × 120 × 2 × 2	3 × 3	✓	
	AvgPool	60 × 120	2 × 2	✓	
rPPG Estimator	Bidirectional LSTM	60 × 120	-	✓	✓
	Linear	60 × 80	-	✓	
	Ordinal	60 × 40	-	✓	
Synthetic Gradient Generator	Conv1DBlock	40 × 120	3 × 3	✓	✓
	Conv1DBlock	20 × 120	3 × 3	✓	✓
	Conv1DBlock	40 × 120	3 × 3	✓	✓
	Conv1DBlock	60 × 120	3 × 3	✓	✓

Source: [27].

**Table 3 sensors-22-09486-t003:** Comparison of EVM network results.

Technique	Dataset	ME	Standard Dev.	RMSE	MAPE	ρ
EVM	MR-NIRP-INDOOR	−8.96	**1.98**	**9.68**	11.46	0.34
EVM	UBFC-rPPG	**2.63**	8.16	15.26	14.28	0.18
EVM with PPG	UBFC-rPPG	8.16	9.12	11.14	**9.74**	**0.52**

**Table 4 sensors-22-09486-t004:** Comparison of Meta-rPPG network results.

Technique	Dataset	ME	Standard Dev.	RMSE	MAPE	ρ
Meta-rPPG	MR-NIRP-INDOOR	196.81	97.16	228.69	406.01	−0.14
Meta-rPPG	UBFC-rPPG	8.93	9.34	12.35	9.56	0.59
Meta-rPPG scalable	UBFC-rPPG	**0.26**	**4.49**	**1.7**	**1.13**	**0.92**
Meta-rPPG scalable	UBFC-rPPG VM	18.55	13.67	23.23	21.54	0.35
Meta-rPPG scalable	UBFC-rPPG SVM	7.38	8.9	11.22	9.29	0.37
Meta-rPPG scalable	UBFC-rPPG GrayScale	27.65	12.75	30.48	30.97	0.74

**Table 5 sensors-22-09486-t005:** Comparison of Meta-rPPG network results from related work.

Technique	Dataset	Standard Dev.	MAE	RMSE	ρ
Meta-rPPG (inductive)	UBFC-rPPG	14.17	13.23	14.63	0.35
Meta-rPPG (proto only)	UBFC-rPPG	9.17	7.82	9.37	0.48
Meta-rPPG (synth only)	UBFC-rPPG	11.92	9.11	11.55	0.42
Meta-rPPG (proto+synth)	UBFC-rPPG	**7.12**	**5.97**	**7.42**	**0.53**

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
