# Peer review of "Remote Heart Rate Prediction in Virtual Reality Head-Mounted Displays Using Machine Learning Techniques"

_sensors, 2022, doi:10.3390/s22239486_

Round 1

Reviewer 1 Report

Remote heart rate prediction in virtual reality head mounted displays using machine learning techniques

Some points need to be known.

  • The picture quality of Figures 1, 2, 3, and 10 could be better. Please improve them.
  • It will be good to highlight the key contributions of the proposed work (in the abstract and the conclusions). 
  • It will be better to list a comparison table to compare the results of machine learning techniques.
  • Moderate English changes are required throughout the manuscript.

Author Response

Dear reviewer, 

Reviewer 2 Report

- Avoid using abbreviations in the abstract as it need to be self contained. 

- Line 36, "Machine learning or Machine Learning (ML) ..." such writing is not necessary. Moreover, the sensors journal and its readership have great affiniting with the computing literature. Thus, the introduction (especially line 35-45) should be deeper than generic superficial facts, but should exhibit the authors knoweldge of the subject. 

- Again, in the introduction, the abbreviation of the techniques is reported without details, which hinders the understanding if the reader is no aware of such highly specific methods. 

- The table of abbreviations, which is required by the template, is missing. 

- Table 2, the results do not make sense. How can the mean average percentage error be greater than 100% in the first line!!!

-  There is no mention of an institutional review board approval or informed consent to include the images of the subjects in the study and the paper. 

- Please double check line 86, the heart rate range, further medical reference should be provided for the maximum heart rate. It does not seem right nor corroborated by an authorized reference. 

- Typo, referencing a figure problem, line 116. 

- The quality of figure 1 should be imrpoved and the proper citation included in the caption (e.g., Feature Image exposed by FirstAuthor et al. [15])

- Section 2 is too long and needs to be more focused .

- The performance metrics need to be defined.

- The numbers in tables 2 and 3, the comma can't be used as a decimal point. 8,93 is equal 893 not 8.93.

Author Response

Dear reviewer, 

Reviewer 3 Report

The authors proposed a machine-learning model for remote heart rate prediction in virtual reality head-mounted displays in this study. Some major concerns are raised as follows:

1. Overall, English language and presentation style should be improved significantly. There were grammatical errors, typos, or jargon.

2. Although the main idea is to build machine learning models, the technical insights of all models were not described clearly. Therefore, the authors should describe this part in detail.

3. When comparing the predictive performance among methods/models, the authors should conduct some statistical tests to see significant differences.

4. The authors only listed some results without in-depth discussions. Therefore, this part should be improved.

5. Machine learning is well-known and has been used in previous studies i.e., PMID: 35767281, PMID: 34989149. Therefore, the authors are suggested to refer to more works in this description to attract a broader readership.

6. Source codes should be provided for replicating the study.

7. Quality of figures should be improved.

8. Some tables should be represented as tables i.e., Figures 2,3.

9. Uncertainties of models should be reported.

Author Response

Dear reviewer, 

Round 2

Reviewer 2 Report

The authors answered all of my comments.

Reviewer 3 Report

My previous comments have been addressed.